# TouchRoller: A Rolling Optical Tactile Sensor for Rapid Assessment of Textures for Large Surface Areas

**DOI:** 10.3390/s23052661

**Published:** 2023-02-28

**Authors:** Guanqun Cao, Jiaqi Jiang, Chen Lu, Daniel Fernandes Gomes, Shan Luo

**Affiliations:** 1Department of Computer Science, University of Liverpool, Liverpool L69 3BX, UK; 2Department of Engineering, King’s College London, London WC2R 2LS, UK

**Keywords:** optical tactile sensor, tactile sensing, tactile perception, robot perception

## Abstract

Tactile sensing is important for robots to perceive the world as it captures the physical
surface properties of the object with which it is in contact and is robust to illumination and colour
variances. However, due to the limited sensing area and the resistance of their fixed surface when they
are applied with relative motions to the object, current tactile sensors have to tap the tactile sensor
on the target object a great number of times when assessing a large surface, i.e., pressing, lifting up,
and shifting to another region. This process is ineffective and time-consuming. It is also undesirable
to drag such sensors as this often damages the sensitive membrane of the sensor or the object. To
address these problems, we propose a roller-based optical tactile sensor named TouchRoller, which
can roll around its centre axis. It maintains being in contact with the assessed surface throughout the
entire motion, allowing for efficient and continuous measurement. Extensive experiments showed
that the TouchRoller sensor can cover a textured surface of 8 cm × 11 cm in a short time of 10 s, much
more effectively than a flat optical tactile sensor (in 196 s). The reconstructed map of the texture from
the collected tactile images has a high Structural Similarity Index (SSIM) of 0.31 on average when
compared with the visual texture. In addition, the contacts on the sensor can be localised with a
low localisation error, 2.63 mm in the centre regions and 7.66 mm on average. The proposed sensor
will enable the fast assessment of large surfaces with high-resolution tactile sensing and the effective
collection of tactile images.

## 1. Introduction

Tactile sensing is one of the key sensing modalities for robots to perceive the world as it conveys important physical properties of the contacted surfaces such as textures, roughness, stiffness, etc. Such tactile information not only enables robots to have a better understanding of the target objects, but also helps facilitate robots’ dexterous manipulation. With the development of tactile sensing, a number of tactile sensors based on different mechanisms have been used extensively in robotic perception and manipulation, such as MEMS, capacitive, strain gauge, and optical tactile sensors [1].

Among various tactile sensors, the camera-based optical tactile sensor has gained great popularity due to its high resolution and low price. Optical tactile sensors use a camera to capture the deformation of a sensitive elastomer on the top of the sensor [2,3], which offers a high spatial resolution with sufficient surface information embedded. However, due to the use of a soft elastomer, which can stick to the contact surface, most of the optical sensors suffer a lack of mobility. The motions of slipping or dragging the sensor on the object’s surface result in unstable contact and a blurred tactile image, and the elastomer of the sensor can be easily damaged by rubbing. In addition, most of the optical tactile sensors are designed for manipulation tasks, and as a result, they have been in the form of a gripper [2] or a finger [4]. Such tactile sensors can only attain local tactile information by physical interaction with a small sensing area. As a result, the evaluation of a large surface using these tactile sensors require numerous contacts with the object, which is ineffective and time-consuming due to repeated data collection movements. It also poses challenges to the design of a tactile sensor that can efficiently collect tactile data.

To facilitate efficient data collection for large surfaces, a tactile sensor is expected to reduce the ineffective movements that the current tactile sensors perform, e.g., pressing on the objects, lifting up, and shifting to another region. Instead, a tactile sensor that can perform effective motions such as rolling or sliding on the object’s surface would be desired. Moreover, as for other tasks such as manipulation, a large sensing area and high sensitivity are also essential for the tactile sensor in the robotic perception of large surfaces.

In this paper, we propose a rolling optical tactile sensor named *TouchRoller*, as shown in Figure 1, which can roll over the surface of an object to collect tactile images continuously. When the sensor come in contact with the object, the elastomer coated on a transparent cylinder distorts to take the geometry of the region with which it is in contact, and the embedded camera inside the cylinder will capture this deformation at a high spatial resolution. A sequence of tactile images of different contact locations will be recorded continuously as the sensor rolls over the target object, which can be used to reconstruct the whole object’s surface effectively. During the rolling motion, the sensor maintains stable contact with the assessed surface, thereby generating clear tactile images and enabling the rapid assessment of large surfaces. To the best of the authors’ knowledge, this work is the very first to present the detailed working mechanism and scientific analysis of a rolling optical tactile sensor to assess a large surface. In our experiments, the TouchRoller sensor demonstrated a good capacity of mapping the textures of the object’s surface. It can cover a textured surface of 8 cm × 11 cm in 10 s, much quicker than the optical tactile sensor can cover a flat surface (in 196 s). In addition, the sensor can localise contacts around the cylinder with a low localisation error, 2.63 mm in the centre regions and 7.66 mm on average.

The contributions of this paper are summarised as follows:A rolling optical tactile sensor that can roll over objects is proposed for rapid assessment of large surfaces;This is the first time a scientific analysis, including the working mechanism, fabrication, geometry, as well as calibration, has been performed for a rolling optical tactile sensor;The experimental results demonstrated that the rolling optical tactile sensor can collect tactile data efficiently with a low localisation error.

## 2. Related Work

In the past few decades, different designs of tactile sensors based on different sensing principles have been proposed, e.g., resistive [5], capacitive [6], piezoelectric [7], and optical tactile sensors [2,3]. Among them, optical-based tactile sensors offer superior sensing resolution [8], compared to other tactile sensors. Tactile array sensors other than optical ones normally contain tens to hundreds of tactile sensing elements. It would be difficult to further increase the amount due to the constraints of the wiring methods. In contrast, optical tactile sensors capture contact information with their built-in cameras, with each pixel taken as a sensing element.

It has been one of the popular methods to use elastic membranes as the tactile skin in the designs of optical tactile sensors, such as the TacTip sensor [3] and the GelSight sensor [2]. The TacTip sensor [3] is an open-source membrane-based tactile sensor inspired by the sensing mechanism of the intermediate ridges in human epidermal layers. The original design [9] had most of its rigid parts 3D-printed despite the manually moulded silicone rubber skin. Arrays of white pins are embedded in the membrane in a way analogous to the intermediate ridges. Any contact that causes the movement of the pins can be detected by the camera. An upgraded version [10] introduced dual-material 3D printing to reduce the manufacturing difficulty and product variance. The TacTip sensor has a simple manufacturing process and reasonable building cost. Recently, the TacTip family [3], including the TacTip, TacTip-GR2, and TacTip-M2, has been developed and used in various applications. In [11], the TacTip sensor was applied to improve the capability of in-hand manipulation for held objects with a low localisation error. In [12], the tactile signal was used to detect the slip between the object and the gripper and react to re-grasping quickly. What is more, object classification and grasp success prediction were performed effectively with the tactile images from the TacTip sensor [13]. However, the measurable area and resolution of the TacTip are limited by the marker-based sensing approach. Furthermore, due to the fixed surface of the sensor, motions such as slipping and dragging will result in unstable contact and may break the membrane.

The GelSight sensor [2] is another type of membrane-based optical tactile sensor. The sensor consists of a transparent elastomer, an optical camera, and a set of LED illumination units. Unlike the TacTip sensor, the transparent elastomer of the GelSight sensor is coated with a reflective surface. As a result, the external illumination will not affect the inside space of the sensor. In a typical sensor–object contact scenario, the surface of the membrane will deform according to the target object and capture its geometry. The first GelSight sensor [14] was created to measure the geometry and texture of the target surface. Following the principle of the prototype, multiple upgraded versions of the GelSight sensor [2,15] have been proposed. Due to the sensitive elastomer and membrane, the GelSight sensor provides the ability to record the textures of the contact surface with fine details and reconstruct the surface height information of the local geometry through the GelSight images. Therefore, the tactile images from the GelSight sensor have been wildly used in material recognition [16], pose change estimation [17], as well as object grasping tasks [18]. However, the earlier designs of the GelSight sensors were bulky, and the sensing area was limited to a small area in the centre of the elastomer. To improve the sensing area, a finger-shaped sensor named the GelTip sensor [4] was proposed, to enable 360-degree contact detection with full resolution.

However, all of these optical tactile sensors suffer from the limitations of small sensing areas and the need for repeated actions such as pressing when assessing large surface areas. In recent years, there have been efforts to improve the mobility of the robot in various tasks, for example the design of a roller-based dexterous hand was proposed in [19,20] to increase the flexibility of in-hand manipulation tasks. The mobility brought by using a roller-based design could also be used to improve the efficiency of assessing large areas with tactile sensing. Compared to a design with a fixed soft membrane relative to the camera in the sensor, a roller-based design for optical tactile sensors can not only improve the mobility of the sensor, but also mitigate the undesirable effects of lateral movements (e.g., slipping and dragging) such as unstable contact and blurred tactile images. A roller-based tactile sensor was mentioned in the review paper [8] on optical tactile sensors, though only a video was referenced instead of a proper publication. To the best of our knowledge, this is the very first work that presents the working mechanism, fabrication, contact localisation, surface mapping, as well as scientific analysis of a roller-based optical tactile sensor.

## 3. The Principle of the TouchRoller Sensor

### 3.1. Overview

As illustrated in Figure 1 and Figure 2, our proposed TouchRoller sensor consists of six main parts: a piece of transparent elastomer; a reflective membrane painted on the elastomer; a transparent cylindrical tube; a camera with a high resolution (640 × 480) to capture the tactile features; embedded LEDs to illuminate the space inside the sensor; as well as supporting parts to connect all the elements of the sensor.

The elastomer layer coated with the reflective membrane covers the surface of the cylinder, where both the cylinder and elastomer are transparent, while the membrane layer is an opaque layer. When the sensor rolls over an object’s surface, e.g., a piece of fabric, the soft elastomer will deform according to the fabric’s surface, and then, the textures and patterns of the fabric will be mapped into this deformation. The cameras will capture the deformation with the help of the reflective membrane and the illumination by the LEDs. As a result, a sequence of tactile images with different locations is collected during rolling, which can be implemented to reconstruct the surface of the whole fabric.

Compared to the other optical tactile sensors discussed in Section 2, we employed a cylindrical shape for the sensor instead of a flat surface, which can roll over the object’s surface to collect the tactile data. The camera is mounted in the centre of the cylindrical tube, and ball bearings are used at the two sides of the sensor to make the camera point towards a fixed angle so that it can capture the tactile features while the sensor is rolling.

### 3.2. Fabrication of the TouchRoller Sensor

In this section, we introduce the fabrication of our sensor, including the elastomer, reflective membrane, and supporting parts of the sensor.

The elastomer, which serves as the contact light medium, is a crucial element in the optical tactile sensor. The elastomer is expected to be soft so that it is able to respond to contact even with a small force, and it must also be optically clear so that the light can go through the elastomer with little refraction and attenuation. Moreover, in order to coat the elastomer on the cylindrical tube, the elastomer is required to be stretchable so that it can closely adhere to the surface of the cylindrical tube. For the fabrication of the elastomer, we made use of the XP-565 from Silicones, Inc., High Point, United State and Slacker from Smooth-On, Inc., Macungie, United State as suggested in [4]. XP-565 Part A, XP-565 Part B, and Slacker were mixed at a ratio of 1:22:22. In this mixture, XP-565 Part B can adjust the softness of the elastomer, while Slacker is used to increase the silicone’s tackiness.

The reflective membrane was used to reflect the light projected by the LEDs with consistent reflectance. The main challenges of making the membrane were the uneven painting, which results in inconsistent reflectance, and cracks caused by the deformation of the membrane, especially on a curved surface. To solve these problems, we mixed the pigment in the dissolved silicone and then painted this mixture on the surface of the elastomer to form the membrane. Specifically, we used the Silver Cast Magic from Smooth-on Company and aluminium powder (mixed at 1:1) as our pigment. Small amounts of XP-565 Part A and Part B, as well as Slacker, the same proportion as for the elastomer, were mixed together with the pigment at first, and then, a silicone solvent was applied to dissolve this mixture to achieve the painting of the membrane. Finally, we performed the painting using this mixture with an airbrush, which can distribute the membrane evenly on the elastomer. As a result, the membrane will obtain a strong toughness and ductility to deform properly with different objects’ surfaces.

The supporting parts, which were used to connect all elements and build up the sensor, consisted of an off-the-shelf transparent glass tube, a set of ball bearings, two steel pipes, two sleeves to fix the glass tube, a supporting plate to place the camera, and the LEDs. As shown in Figure 2, the steel pipes were connected to the supporting plate, and the pipes went through the sleeves, while the sleeves were used to fix the glass tube. Moreover, the ball bearings were implemented between the pipes and the sleeve to enable the supporting plate to point at a fixed angle while the sensor rolls over an object’s surface. Finally, we placed the camera, with a resolution of 640×480 and a 2.8 mm focal length, on the supporting plate, and a set of LEDs was placed beside the camera to illuminate the space inside the sensor. In this work, we used the 3D printer Anycubic i3 Mega to make the customised parts, including the sleeves and the supporting plate. Overall, our TouchRoller sensor had a weight of 420 grams, with a diameter of 100 mm and a length of 100 mm. All the 3D-printed models can be found at https://github.com/3PTelephant/TouchRoller (accessed on 27 February 2023).

### 3.3. Surface Projection

In robotic inspection, it is crucial for the robot to know the location of the target object by tactile feedback. Different from the GelSight sensor with a flat surface, our proposed rolling tactile sensor has a curved surface, and it is necessary to determine the contact location from the tactile image collected by our sensor. To this end, we derived the location of the contact on the surface of our sensor, given the contact pixels detected in the tactile image.

As illustrated in Figure 3, we modelled the surface of our TouchRoller sensor as a cylinder with radius *r*. The *x*-axis coincides with the central axis of the cylinder, and the *z*-axis is vertically downward, with the referential origin (0,0,0) placed at the centre of the cylinder. Therefore, the surface point (x,y,z) of the sensor satisfies the following equation:(1)$$y^{2}+z^{2}=r^{2}$$

We modelled the camera as a pinhole camera model oriented along the *z*-axis, with the optical centre of the lens placed at the referential origin. The transformation of a contact point $P={\left[X_{w},Y_{w},Z_{w}\right]}^{T}$ to a pixel $P^{\prime }={\left[u,v\right]}^{T}$ in the tactile image can be represented [21] as:(2)$$\lambda \left[\begin{array}{c} u\\ v\\ 1 \end{array}\right]=K\left[\begin{array}{cc} R & t\\ 0 & 1 \end{array}\right]\left[\begin{array}{c} X_{w}\\ Y_{w}\\ Z_{w}\\ 1 \end{array}\right]$$
where both *P* and $P^{\prime }$ are denoted in homogeneous coordinates, λ is a scale factor for the image point, *K* is the matrix of the intrinsic parameters of the camera, and *R* and *t* represent the rotation and translation of the camera, respectively. Firstly, *P* is transformed by the camera’s extrinsic matrix, which gives the transformation between the camera coordinates and the world coordinates, and then mapped by the intrinsic matrix *K* to obtain the pixel in the tactile image.

In the camera’s intrinsic matrix *K* [22]:(3)$$K=\left[\begin{array}{cccc} \frac{f}{d_{x}} & 0 & u_{0} & 0\\ 0 & \frac{f}{d_{y}} & v_{0} & 0\\ 0 & 0 & 1 & 0 \end{array}\right]$$
*f* represents the focal length of the camera; $d_{x}$ and $d_{y}$ denote the pixel size; $\left(u_{0},v_{0}\right)$ is the centre point of the tactile image.

When the tactile sensor rolls over a surface, the camera is expected to be oriented along the *z*-axis with the optical centre of the lens at the centre of the cylinder to capture the deformation of the membrane. However, the camera may rotate while rolling, and the location of the camera may change due to the rotation. Considering that the error in other orientations was very small, which can be ignored in our sensor, we only focused on the rotation about the *x*-axis and the translation on the *z* axis in the extrinsic matrix. Specifically, *R* and *t* can be represented as follows [22]:(4)$$R=\left[\begin{array}{ccc} 1 & 0 & 0\\ 0 & \cos\theta & \sin\theta \\ 0 & -\sin\theta & \cos\theta \end{array}\right]$$
(5)$$t={\left[0,0,d\right]}^{T}$$
where θ represents the rotation angle and *d* denotes the length of the vertical translation. By combining all the above equations, we can obtain
(6)$$\left\{\begin{array}{c} \lambda u=X_{w}f_{x}-Y_{w}u_{0}\sin\theta +Z_{w}u_{0}\cos\theta +u_{0}d\\ \lambda v=Y_{w}\left(f_{y}\cos\theta -v_{0}\sin\theta \right)+\\ \;\;Z_{w}\left(f_{y}\sin\theta +v_{0}\cos\theta \right)+v_{0}d\\ \lambda =-Y_{w}\sin\theta +Z_{w}\cos\theta +d\\ r^{2}=Y_{w}^{2}+Z_{w}^{2} \end{array}\right.$$
where the normalised focal length $f_{x}$ and $f_{y}$ denote $\frac{f}{d_{x}}$ and $\frac{f}{d_{y}}$, respectively. As a result, we can calculate the surface point $P={\left[X_{w},Y_{w},Z_{w}\right]}^{T}$ according to the known $P^{\prime }={\left[u,v\right]}^{T}$ in the tactile image using Equation (Equation 6).

### 3.4. Calibration

Similar to other optical tactile sensors, there is an inevitable offset of the camera axis from the centre line of the sensor shell, and the offset can be exaggerated by the resolution of the 3D printing of the shell. To rectify the offset, the TouchRoller needs to be calibrated to improve the accuracy of the sensor, especially in the surface projection tasks described in Section 3.3. The calibration consisted of determining both the intrinsic camera parameters and the transformation matrix, which maps between the camera coordinates and the world coordinates.

Firstly, the intrinsic parameter of this camera was estimated with the widely used camera calibration method [22], as shown in Figure 4a. Then, we used a small 3D-printed rectangle frame, as illustrated in Figure 4b, which consisted of 2×5 hemispheres with a radius of 1 mm, to estimate θ and *d* in Equation (Equation 6). During the calibration, we placed our sensor on the surface of the rectangular frame and recorded a set of tactile images, with one example shown in Figure 4c.

By using multiple image processing methods including binarisation and morphological filtering, we can obtain the centre of those hemispheres in the image space. Moreover, we define a reference coordinate in the rectangle frame, as shown in Figure 4c. With the known size of the frame and detected hemispheres centres, we first used the Iterative PnP solver [23] to calculate the transformation matrix between the camera coordinates and the reference coordinates by minimising the reprojection error in the pixel plane and then obtained the rotation angle $\theta_{i}$ and translation error $t_{i}$ for each tactile image. Finally, we determined θ and *t* with the average of all estimated $\theta_{i}$ and $t_{i}$.

During the use of the sensor, there are several aspects that can affect the accuracy of the measurement. For instance, the elastomer may wear out because of frequent contact. However, the elastomer can be easily substituted with a new one. What is more, the dirt accumulation of the sensor, either inside the sensor or on the membrane, may lead to undesirable effects on the collected tactile data. One potential way to solve this problem is to clean the interior and exterior of the device after disassembling it and re-calibrate the sensor after it is re-assembled.

## 4. Experiments and Analysis

In this section, we report the results obtained from the set of experiments carried out to validate the design of our sensor. To validate that our proposed sensor is able to assess the textures of large surfaces, a surface mapping experiment was conducted, where the whole surface texture was reconstructed using the tactile data obtained from various locations with a single linear rolling motion. Additionally, the capability to locate a contact region is crucial in robotic manipulation and exploration, which allows the robot to actively perceive objects. Hence, the contact localisation experiment was performed to derive the contact location in the world coordinates from a tactile image. Finally, to illustrate the efficiency of our proposed sensor in tactile data collection, we compared the time consumption of the data collection using our proposed sensor against the GelSight sensor, which has a flat surface.

### 4.1. Surface Mapping

We experimented with mapping the texture of an 8 cm × 11 cm piece of fabric using three different speeds: slow (15 s), medium (10 s), and fast (5 s). As shown in Figure 1, a video was captured by the internal camera of the sensor as it rolled over the textured fabric for one full revolution at 30 FPS. Thin centred patches corresponding to the contacted region, with a height of 70 pixels, were then cropped from the captured video frames and stitched into one single tactile map. Each patch can be considered approximately flat, orthogonal to the camera sensor and, due to the cylindrical design of the sensor, all captured at the same distance, *r*.

As such, stitching the entire map required only the finding of the Δy translations, for every pair of consecutive frames, which minimises the Mean Absolute Error (MAE) between two patches in the pair. Δy is searched in [−25,25]. As expected, as shown in Figure 5 (the negative sign is representative of the direction of the motion, i.e., bottom-up), |Δy| is directly proportional to the speed of the sensor.

For the quantitative analysis, the obtained map was compared with a grey-scale top-down view of the same textured fabric. To minimise the differences due to the two cameras’ viewing angles and the distances to the surface and horizontal alignment, two pairs of points in the real and virtual frames were selected and constrained to fall into the two corresponding vertical lines. A third pair of points was also derived such that the three points in each image formed a right isosceles triangle. The OpenCV getAffineTransform and warpAffine functions were then applied to find the affine transformation between the tactile map and the grey-scale top-down view. This constrained alignment ensured that the spatial proportionality of the generated tactile map was preserved.

Three evaluation metrics were chosen to assess the quality of the obtained tactile map: Structural Similarity (SSIM) [24], Peak-Signal-to-Noise Ratio (PSNR), and Mean Absolute Error (MAE). However, as shown in Table 1, the metrics failed to capture the poorest quality of the fast mapping, and the failure may come from the difference between the visual and tactile modalities. As shown in Figure 6, while the fast motion clearly showed signs of motion blur, the medium and slow motions appeared well focused. As such, from this experiment, it can be concluded that as long modest velocities are used (around 1.1 cm/s), the proposed *TouchRoller* sensor can be used to assess large surfaces rapidly.

### 4.2. Contact Localisation

Apart from the data collection, it is also important for the robot to understand the contact location so that the robot is able to adjust its pose to explore the object actively. To this end, we used a 3D-printed solid to tap on the surface of our sensor with different positions and derive the location in the world coordinates using the tactile image from our sensor. As a result, the localisation error can be measured between the derived location and the real contact location of the object.

As illustrated in Figure 7a, a small solid with three sticks was printed with different heights. When we placed the solid vertical to *x*-axis from both the front and back sides, the specific heights corresponded to different central angles between the vertical direction and the connection of contacted points with the centre of the circle (a certain section of the cylinder), i.e., −π/6,−π/12,0,π/12,π/6, respectively. As shown in Figure 7b–d, the solid was also tapped on the sensor along the *x*-axis at different locations, i.e, 1/4,1/2,3/4 length of the sensor.

To determine the contact location in the tactile image, we designed an extraction method. Firstly, the tactile image including the contact regions was grey-scaled and Gaussian blurred, then binarised using a proper threshold. Next, we implemented the FindContours function from OpenCV on the binarised image to determine the contact regions. Finally, we made use of the moments function from OpenCV to calculate the central points of each contact region, and these central points represent the contact locations in the tactile images.

As we calculated the contact point in the tactile image, we were able to estimate the contact location using the surface projection method (detailed in Section 3.3). The sensor was first calibrated after being mounted onto the robot arm, and then, the intrinsic matrix *K* and extrinsic matrix, which contain *R* and *t*, can be obtained. Since the radius of the cylinder *r* is known, the contact location $P={\left[X_{w},Y_{w},Z_{w}\right]}^{T}$ can be estimated using Equation (Equation 6), with the detected contact point $P^{\prime }={\left[u,v\right]}^{T}$ in the tactile image.

As shown in Table 2, the Euclidean distances between the estimated and the real contact location were computed, including both the mean and the standard deviation of the distance. We can find that the centre of the sensing area has the smallest error, around 2.63 mm, compared with other regions. The error became larger as the sensing area goes further from the centre. The error may arise from multiple factors: the calibration of the camera had errors; the detection of the contact point in the tactile image was not precise enough; the thickness of the elastomer was uneven, and therefore, the sensor was not a perfect cylinder.

### 4.3. Comparison with Other Optical Tactile Sensors

In this section, our proposed sensor is compared with an optical tactile sensor with a flat surface, i.e., the GelSight sensor, in terms of the time consumption and the definition of the tactile features during the data collection. The GelSight sensor always presses against the object with a flat sensing area. If we slide the GelSight over an object, the tactile image will be blurred due to the unstable contact (shown in Figure 8a). Hence, the data collection by the GelSight requires a series of actions: pressing, lifting up, shifting to another region, and repeating again, which are very time-consuming. In contrast, our proposed sensor is able to roll over the object with stable contacts, and it can generate a clear tactile image (shown in Figure 8b) with a rapid assessment of large surfaces. Specifically, to collect the tactile textures of a piece of fabric of 8 cm × 11 cm, the GelSight sensor, with a sensing area of 1.6 cm × 1.2 cm, requires at least 49 pressing actions to cover the fabric, while the TouchRoller only needs to roll over the fabric once. To evaluate the time efficiency of our proposed sensor while assessing large surface areas, we attached our TouchRoller sensor as the end-effector on the UR5 robot to collect the tactile textures. Concretely, the fabric was placed on a flat plane, and the UR5 was controlled to press the sensor on the fabric with about 10 N of force, resulting in clear textures, and roll over the fabric at a constant horizontal speed of around 1.1 cm/s. If we compare the time consumption against using the GelSight sensor with a similar motion speed, the GelSight sensor took about 196 s, while the TouchRoller only took 10 s, which promoted the sampling efficiency significantly.

## 5. Conclusions

In this paper, we introduced the design for a rolling optical tactile sensor, *TouchRoller*, which has a cylindrical shape and can roll over an object’s surface to improve the efficiency of data collection. It demonstrated the ability of a faster assessment of large surfaces by rolling the sensor over the surface of objects compared with other optical tactile sensors. Our experiments showed that the sensor can be used to reconstruct the surface of an object effectively and the contact on the sensor surface can be localised with high accuracy. Our TouchRoller tactile sensor can be applied in fast surface inspection including crack detection in structural health monitoring in the future.

On the other hand, as a tactile sensor for robotic perception, our proposed TouchRoller has some limitations. It is noticeable that the proposed sensor is large in size, which makes it not suitable to collect data in a limited space or measure curved surfaces. Additionally, its degrees of freedom are limited to the end-effector of the robot arm, which makes it challenging to utilise such a sensor for dexterous manipulation tasks. Moreover, the soft elastomer covering the surface of the sensor is delicate and is prone to wear out by frequent contact or touching a surface with a high temperature. Finally, the proposed sensor needs regular re-calibration and maintenance, which could be costly and time-consuming. In future work, we will attempt to optimise the fabrication process and create durable materials for the sensor. We also plan to reduce the size of the sensor to make it easier to be mounted onto robot grippers so that it can be applied in robotic manipulation tasks.

## Figures and Tables

**Figure 1 sensors-23-02661-f001:**
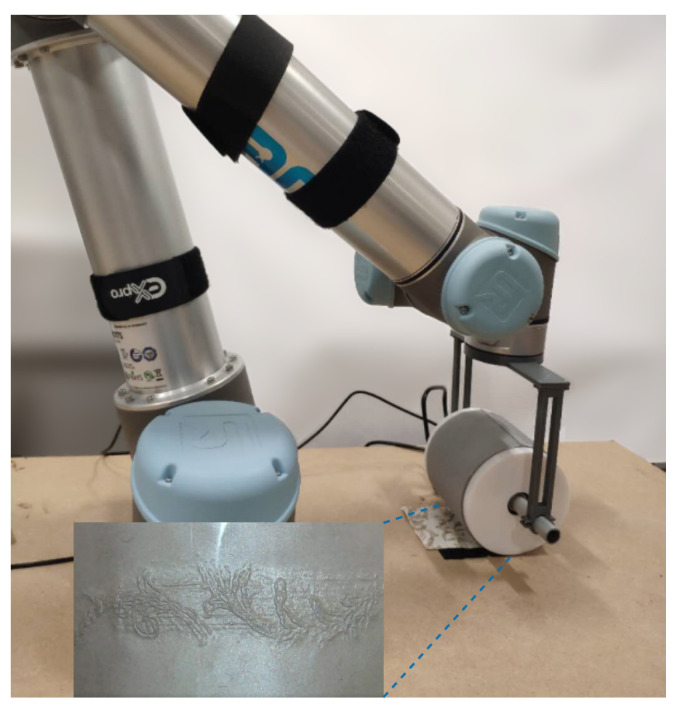
Our TouchRoller sensor mounted on the end of a UR5 robot arm is rolling over a piece of fabric to collect the surface texture, and the collected tactile image of the fabric texture is shown on the bottom left.

**Figure 2 sensors-23-02661-f002:**
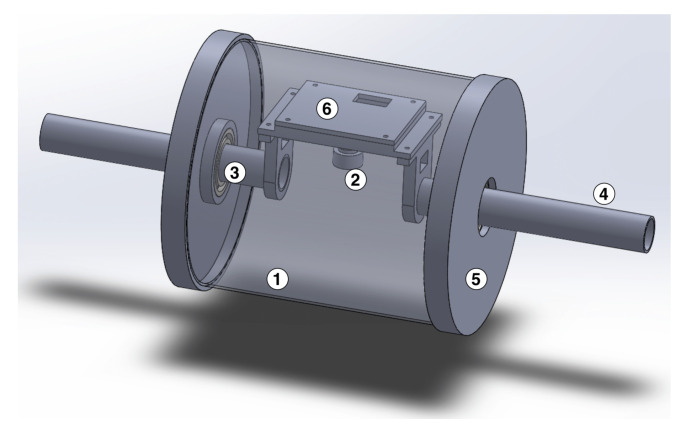
The design of the TouchRoller tactile sensor. (1) The transparent cylindrical tube; (2) the webcam; (3) ball bearings to make sure the camera point towards the region in contact with the object; (4) steel pipes; (5) sleeves to fix the cylindrical tubes and pipes; (6) the supporting plate to place the camera.

**Figure 3 sensors-23-02661-f003:**
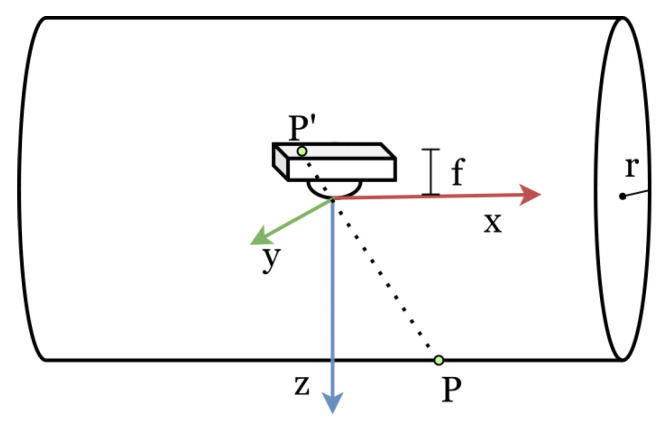
The geometrical model of the TouchRoller tactile sensor. The sensor was modelled as a cylinder with a radius of *r*. A camera with focal length *f* was placed in the centre of the cylinder, and a point *P* on the surface of the cylinder was mapped to $P^{\prime }$ in the tactile image.

**Figure 4 sensors-23-02661-f004:**
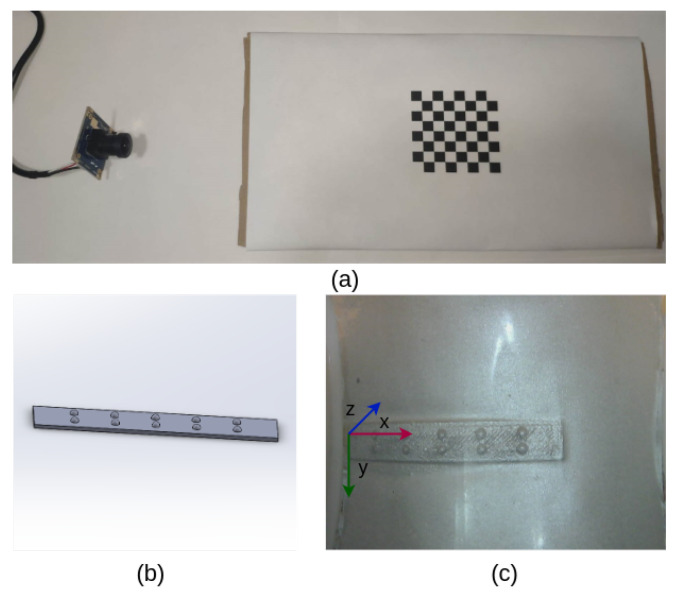
The calibration of the TouchRoller sensor. (**a**) A 7×7 chessboard was used to calculate the intrinsic matrix of the camera. (**b**) A 3D-printed solid with an array of 2×5 hemispheres was employed to estimate *R* and *t*. (**c**) The corresponding tactile image when the sensor presses on the 3D-printed solid shown in (**b**).

**Figure 5 sensors-23-02661-f005:**
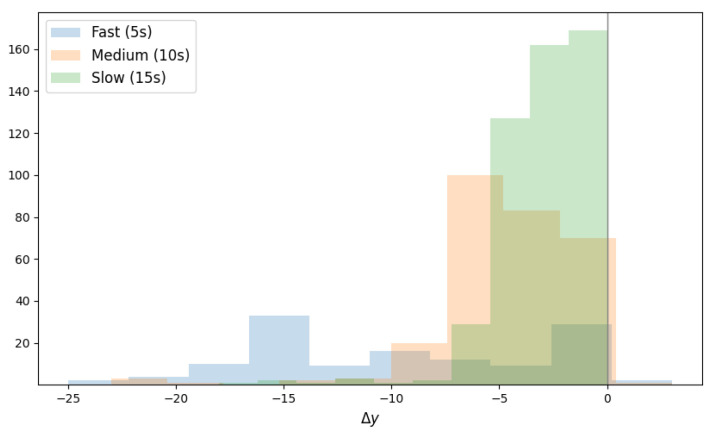
Histograms of the chosen Δy for the three motion speeds.

**Figure 6 sensors-23-02661-f006:**
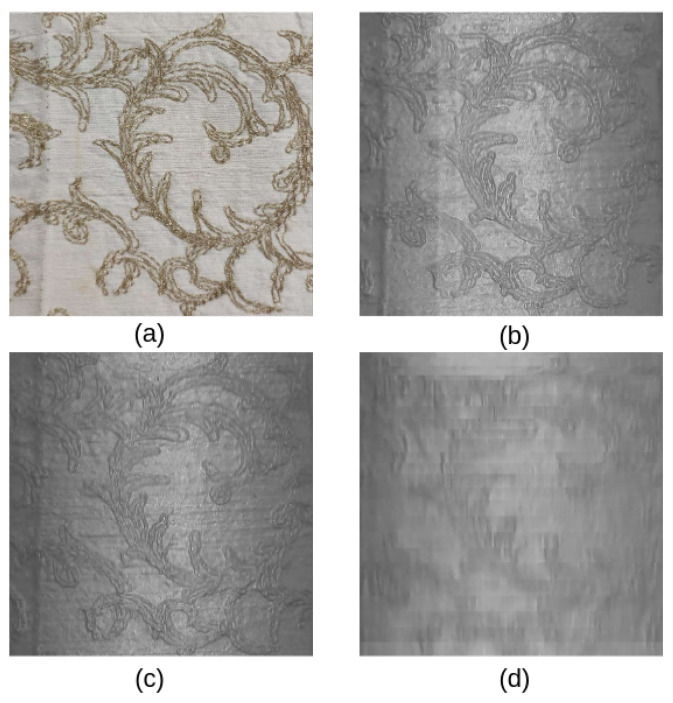
Mapped surface. (**a**) The visual top-down view and (**b**–**d**) representative samples of the tactile maps obtained with the corresponding slow (15 s), medium (10 s), and fast (5 s) motions.

**Figure 7 sensors-23-02661-f007:**
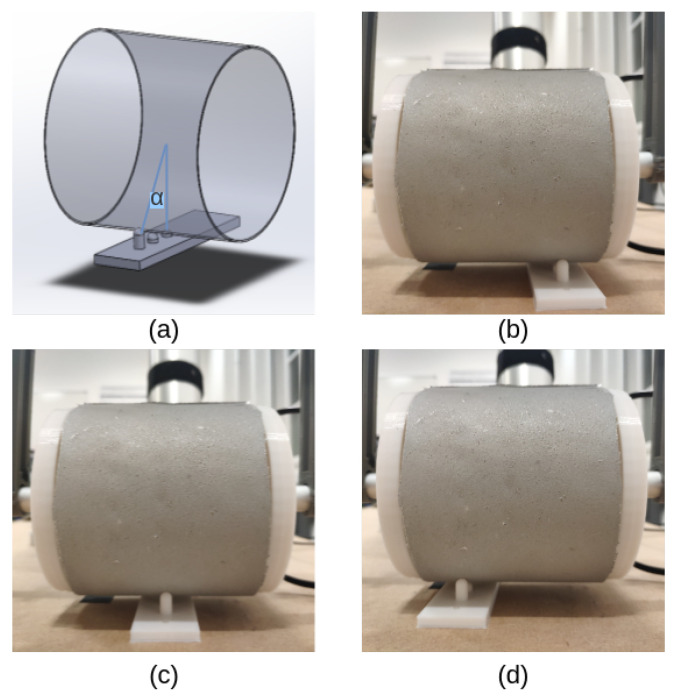
Contact localisation. (**a**) A 3D printed solid with three sticks of different heights is tapped vertically to the sensor (α is the central angles according to the heights). (**b**–**d**) The solid is also tapped at different locations of the sensor.

**Figure 8 sensors-23-02661-f008:**
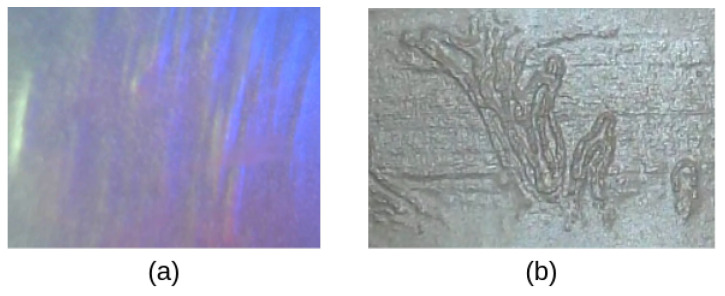
(**a**) The blurred tactile image collected by the GelSight while slipping on the fabric. (**b**) The clear tactile image by the TouchRoller while rolling over the same piece of fabric.

**Table 1 sensors-23-02661-t001:** Comparison of the captured tactile map with the corresponding textured fabric top-view, for the three motion velocities: slow (15 s), medium (10 s), and fast (5 s).

	SSIM	PSNR	MAE
Slow (15 s)	0.291±0.010	15.09±0.13	15.03±0.28%
Medium (10 s)	0.305±0.010	14.41±0.71	16.43±1.51%
Fast (5 s)	0.330±0.007	15.59±0.94	14.04±2.04%

**Table 2 sensors-23-02661-t002:** Euclidean distances between the estimated and the real contact locations (expressed in millimetres).

Location	1/4 Length	1/2 Length	3/4 Length
−π/6	9.64±0.09	11.13±0.06	13.75±1.89
−π/12	7.53±0.19	4.50±0.08	8.26±1.46
0	5.00±0.59	2.63±0.74	6.89±0.79
π/12	6.42±0.32	4.06±0.26	6.66±0.53
π/6	8.83±0.10	8.58±0.19	12.34±0.21

## Data Availability

The datasets applied in the current study are available from the corresponding author upon reasonable request.

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
