# Peer review of "TouchRoller: A Rolling Optical Tactile Sensor for Rapid Assessment of Textures for Large Surface Areas"

_sensors, 2023, doi:10.3390/s23052661_

Round 1

Reviewer 1 Report

sensors-2230606

TouchRoller: A Rolling Optical Tactile Sensor for Rapid Assessment of Large Surfaces

To overcome the disadvantages of ineffective and time-consuming of traditional optical tactile sensor, a TouchRoller is designed for the fast assessment physical properties of large surfaces. The performance of the present measurement sensor is checked by experiments.  However, I propose several questions for the author to explain, so as to effectively improve the quality of this manuscript. Specific comments are as follows:

1. Title must be carefully revised for the contents of assessment are not given.

2. It is mentioned in the manuscript that the experiments are the key of this manuscript to check the performance of the present TouchRoller, but I did not see clearly the verification instructions specifically about this in the experimental part of the manuscript. So, explanations of the experiments should be carefully given. 

Reviewer 2 Report

The paper reports on a realisation of a tactile sensor using a camera for robotic applications. I support the publication of this paper but I have several comments that I would suggest that the authors address:

1.

Abstract: what is (in196s)?

2. 

The authors write:

"The sensor calibration aims to find both intrinsic parameter of this camera and the exact 204 transformation matrix between the camera coordinate and the world coordinates. From Sec 3.3, 205 we can map the pixel positions in the TouchRoller images to the world coordinates. However, the 206 camera could not be absolutely vertical down and centred in the tube due to the error of installation 207 and construction. Hence, the calibration is mandatory for precise measurement"

This text makes an impression that the sensor has not been assembled accurately (camera has and error of installation) and that the authors should work more on the sensor to remove the problem first (install the camera precisely) before carrying out the study and publishing the results. Is this true? If so then I think that this issue should be rectified before publishing results. If not, then the authors should word their description more carefully and more clearly explain what they mean. Is this problem intrinsic and not removable no matter how much time one spends aligning? If so why? This needs to be make clear to the reader.

3.

Do the authors know at which wavelengths the cylinder is transparent. Would the system benefit from some additional illumination? I understand that at the moment the system relies on the ambient light to work properly? So, if the light is switched off the system would not work? Would the system work better if the cylinder transparency was improved? Does dirt accumulation reduce the effectiveness of the system? How one could reduce the effects of dirt accumulation?

4.

Fig.5 Is possible to parametrise the recorder histogram to a known probability distribution? Is it maybe a binomial distribution? Could the authors check a longer time also to make sure that 15 s is sufficient to achieve maximum accuracy?  

Reviewer 3 Report

This article deals with developments in the field of robotic systems, mainly in the field of object grasping sensing. Tactile sensing is an interesting approach for robots to perceive objects that need to be manipulated. What is needed is an effective tool for sensing and grasping objects, which will enable detection and grasping of objects in a short time. Improving the efficiency of tactile sensing is one of the ways to advance the development of robotic systems and prepare a progressive tool for time-saving sensing and manipulation of objects. Therefore, this problem needs to be solved, and this article solves a current issue that will be interesting for readers and people from industrial practice, where these systems can find application.
To solve this application task, the authors designed a unique roller optical tactile sensor called TouchRoller, which can rotate around its central axis. The proposed scanning system will enable quick scanning of large surfaces using tactile sensing and evaluation of such surfaces of manipulated objects. TouchRoller sensor is mounted on the end of UR5 robot arm for experimental verification of the designed system. The TouchRoller rolls over the scanned surface continuously to obtain a tactile image. At the moment when the TouchRoller comes into contact with an object, the elastomer placed on the contact translucent cylinder changes its geometry upon contact with the regions on the scanned surface and this is detected by a camera located inside this cylinder.
This principle could also be used for tactile sensing of larger areas.
The clear benefit and contribution of this article lies in the development of a new tactile sensing methodology that can be used in the field of robotics.
At the beginning of the article, there is an overview of the current situation in this area, and the authors also explain the reasons for solving this issue. Below is an overview of similar works in this area.
Surface projection math description is described in the next part of this article.
Calibration and data processing of the designed system are other problems that the authors successfully solved.
The experiments showed a relatively good ability to map the detected surface of the object in a relatively short time.
It is also useful that the authors compared their designed system with other optical tactile sensors.
The conclusion summarizes the findings from this research and also states how the authors will continue to solve this issue.
This article is properly prepared and the excellent thing is that it is practically oriented and there are also experiments to confirm the theoretical assumptions.

The article is prepared perfectly, but there are several formal errors in the article and I have some ambiguities that need to be explained in the article.

Comments:
on lines 10, 54, 296 are listed "8cm × 11cm" units are as italic style. Units should be listed as normal text.
It is also bad on lines 164 "2.8mm" and then also on line 168 "100mm". Also on line 282 "2.63mm" is wrong. It is also bad further on in the article. It needs to be checked
In the article, I did not find a description of the methodology of how the UR5 robot was guided to follow the evaluated area. Please describe the procedure as it was implemented.
In the conclusion, it is mentioned that this system has its limitations. Please specify in more detail what the limitations and weaknesses of this system are.
What is the repeatability of this method in the case of multiple evaluations of the same surface using this method.
How the speed of data processing depends on the size of the evaluated area. What is more advantageous? Create a smaller tactile sensing system or a larger one in terms of application speed and efficiency.
How is the service life of such a system with its long-term use? How often is it necessary to perform the calibration of this system.

Reviewer 4 Report

The TouchRoller optical tactile sensor, which can roll about its central axis, is proposed in this work. Throughout the whole motion, it remains in touch with the surface being measured, enabling effective and continuous measurement. An 8 cm by 11 cm textured surface can be covered by the proposed sensor in just 10 seconds, which is a significant improvement over a flat optical tactile sensor, according to experiments. When compared to the visual texture, the reconstructed map of the texture from the gathered tactile pictures has a high Structural Similarity Index of 0.31 on average. The contacts on the sensor may also be localized with a minimal localization error, according to the authors, which is 2.63 mm in the central areas and 7.66 mm on average. The paper has been well  written and can be accepted after a minor revision:

-check the paper for some writing errors; for example see line 119

-the basic equation 3-6 need references

-a comparison with a similar techniques will be helful
